# Impact of COVID-19 restrictions on preschool children's eating, activity and sleep behaviours: a qualitative study

Joanne Clarke ,[1] Ruth Kipping,[2] Stephanie Chambers,[3] Kate Willis,[2] Hilary Taylor,[2] Rachel Brophy,[2] Kimberly Hannam,[2] Sharon Anne Simpson,[3] Rebecca Langford [2]

¹Institute of Applied Health Research, University of Birmingham, Birmingham, UK
²Centre for Public Health, Population Health Sciences, Bristol Medical School, University of Bristol, Bristol, UK
³School of Social and Political Sciences and MRC/CSO Social and Public Health Sciences Unit, Institute of Health and Wellbeing, University of Glasgow, Glasgow, UK

**Correspondence to**
Dr Rebecca Langford;
Beki.langford@bristol.ac.uk

## ABSTRACT

**Objectives** In spring 2020, the first COVID-19 national lockdown placed unprecedented restrictions on the behaviour and movements of the UK population. Citizens were ordered to 'stay at home', only allowed to leave their houses to buy essential supplies, attend medical appointments or exercise once a day. We explored how lockdown and its subsequent easing changed young children's everyday activities, eating and sleep habits to gain insight into the impact for health and well-being.

**Design** In-depth qualitative interviews; data analysed using thematic analysis.

**Setting** South West and West Midlands of England.

**Participants** Twenty parents (16 mothers; 4 fathers) of preschool-age children (3–5 years) due to start school in September 2020. Forty per cent of the sample were from Black, Asian or minority ethnic backgrounds and half lived in the most deprived areas.

**Results** Children's activity, screen time, eating and sleep routines had been disrupted. Parents reported children ate more snacks, but families also spent more time preparing meals and eating together. Most parents reported a reduction in their children's physical activity and an increase in screen time, which some linked to difficulties in getting their child to sleep. Parents sometimes expressed guilt about changes in activity, screen time and snacking over lockdown. Most felt these changes would be temporary, though others worried about re-establishing healthy routines.

**Conclusions** Parents reported that lockdown negatively impacted on preschool children's eating, activity and sleep routines. While some positive changes were identified, many participants described lack of routines, habits and boundaries which may have been detrimental for child health and development. Guidance and support for families during COVID-19 restrictions could be valuable to help maintain healthy activity, eating, screen time and sleeping routines to protect child health and ensure unhealthy habits are not adopted.

## STRENGTHS AND LIMITATIONS OF THIS STUDY

⇒ Few studies have explored the perceived impact of COVID-19 restrictions on young children or health behaviours like physical activity or diet.

⇒ Our in-depth interviews provide novel insights into the perceived impact of lockdown restrictions on preschool children's physical activity, sedentary behaviour, food intake and sleep.

⇒ A strength of the study was the ethnic and socio-economic diversity of the sample with half of participants from the lowest deprivation quintile and 40% from Black, Asian or minority ethnicities.

⇒ Interviews were conducted with parents, rather than preschool children themselves, for pragmatic reasons.

limit the spread of the virus, including social distancing and local or national 'lockdowns'.

The UK government imposed its first 'lockdown' in March 2020. Citizens were required to 'stay at home', only leaving to buy essential supplies, attend medical appointments or to exercise. Schools and nurseries were closed (except to vulnerable or 'keyworker' children), as were leisure facilities, pubs, restaurants, theatres and non-essential shops. Non-essential workplaces were closed, and people could not meet anyone beyond their immediate household. Unlike more stringent lockdowns elsewhere, UK citizens were permitted to exercise outside once a day.[2] Restrictions lasted over 2 months until late May 2020. Preschools reopened to all children in June 2020, alongside schools who reopened to certain year groups at this point; however, occupancy remained low with attendance half that recorded for the previous year (37% vs 77%).[3]

Despite being least affected by the virus[4] children have experienced enormous disruption which may have impacted their health behaviours. Children under 5 should have 180 min of physical activity a day.[5] Children

## BACKGROUND

The emergence of COVID-19, and subsequent efforts to restrict its spread, led to unprecedented economic and social disruption.[1] Many countries imposed restrictions on citizens' behaviours and movements to

aged 3–4 should have 11–13 hours of sleep in a 24-hour period.[6] The closure of preschools, playgrounds and the 'stay at home' order during lockdown reduced children's opportunities for physical activity, increased the likelihood of sedentary behaviours[7] and had the potential to disrupt sleep patterns.[8] The spring 2020 lockdown also changed people's food shopping and eating habits. Consumers moved to less frequent shopping trips, but spent more on food than prelockdown, potentially changing the home food environment.[9]

Establishing healthy behaviours in the early years is important in maintaining a healthy weight.[10] Nearly one in four UK children starting primary school are overweight or obese, with rates increasing with deprivation.[11] Many children saw time in childcare reduced or stopped altogether.[12] Preschool children require greater parental supervision which may mean managing children's activities and behaviours in lockdown was more challenging for these families, particularly if parents were working or there were additional family stressors.[13]

Several surveys have sought to capture the impact of COVID-19 on children and young people's health and well-being. Most have focused on emotional well-being and have often targeted older children.[14] Less attention has focused on younger (preschool) children or explored the impact on physical activity, sedentary behaviour and diet. An exception is the Covid-19: Supporting Parents and Young Children during Epidemics (Co-SPYCE) study[15] which surveyed caregivers of 2–4 year-olds. It found 26% of children had 3+ hours of screen time a day, and only 22% were active for 180 min/day. Public Health Scotland developed the COVID-19 Early Years Resilience and Impact Survey[16] for 2–7 year-olds in Scotland. From 11 228 participants, eating behaviour was identified as worse by 32% of parents. There were mixed responses about physical activity, with 24% reporting children doing more during lockdown and 47% reported less.

While these quantitative surveys provide a useful overview, few studies have used qualitative methods to understand in greater depth how preschool children have been affected by these restrictions. To address this gap, we used in-depth interviews with parents of 3–5 year-olds to explore the impact of the first UK COVID-19 lockdown on preschool children's eating, activity and sleep behaviours.

## METHODS

### Population, sampling and recruitment

Criteria for inclusion were: (1) parent/carer of a child aged 3–5 years in their final year at preschool; (2) child usually attends preschool at least once per week; and (3) child due to start school in September 2020. Non-English speakers were excluded for pragmatic reasons.

Parents were recruited via two methods. First, we asked nursery staff participating in another research study[17] to disseminate study information to parents via email or social media. This included nurseries from Swindon and Somerset (South West England) and Sandwell (West

---

**Box 1  Summary of topic guide used by interviewers to guide discussions during semistructured interviews with parents of preschool children. The full topic guide is available as online supplemental materials. This was created by the coauthors for this publication**

Interview topics

*Physical activity*
⇒ Impact on activity in and out of house. More/less/same as before? Why?

*Sedentary activities*
⇒ Impact on and types of activities. More/less/same as before? Why?

*Sleep behaviours*
⇒ Impact on bedtimes, getting to sleep and staying asleep. More/less/same as before? Why?

*Eating*
⇒ Impact on what and when food is eaten (mealtimes and snacks). More/less/same as before? Why?

*Food purchasing*
⇒ Amount, type, cost and access to food.

*Food preparation*
⇒ Changes to who, how, when food is prepared and why.

---

Midlands). Second, we posted study information on local Facebook groups within these areas and Birmingham (West Midlands) to increase recruitment of Black, Asian and minority ethnic (BAME) participants. Interested parents (n=85) completed a form to check eligibility and collect postcode (deprivation) and ethnicity data for sampling purposes. To provide diversity, sampling was weighted towards parents in the most deprived areas and those from BAME populations. Twelve parents were recruited via nurseries and eight via Facebook.

Sampled parents were emailed a participant information sheet and consent form. Interviews were arranged with participants at least 24 hours after sending the study documents. All sampled parents agreed to participate, bar one who we were subsequently unable to contact.

### Patient and public involvement

Due to time constraints, we did not directly involve parents or children in the planning of this study. However, nursery staff provided advice on the best ways to recruit parents for this study.

### Data collection

Semistructured interviews were conducted by experienced qualitative researchers (JC, KW or SC) in July/August 2020 by telephone (n=18) or video (n=2). Before interview, the researcher explained the study aims, answered questions and gained consent. A topic guide (box 1, see also online supplemental materials) was used to guide discussions with parents encouraged to talk openly about their experiences. Interviews were audio recorded, transcribed verbatim and anonymised. Following each interview, notes were made of key points raised. A £30 shopping voucher was sent to participants following the interview.

**Table 1** Demographic characteristics of the parents of preschool children who participated in a semistructured interview (n=20)

| Characteristic categories | | n (%) |
|---|---|---|
| Gender | Female | 16 (80) |
| | Male | 4 (20) |
| Age group | 21–25 | 1 (5) |
| | 26–30 | 7 (35) |
| | 31–35 | 5 (25) |
| | 36–40 | 2 (10) |
| | 41–45 | 5 (25) |
| Ethnicity | White British or White Other | 12 (60) |
| | Asian or Asian British | 4 (20) |
| | Black or Black British | 1 (5) |
| | Mixed | 3 (15) |
| IMD quintile (1=most deprived) | 1 | 10 (50) |
| | 2 | 2 (10) |
| | 3 | 3 (15) |
| | 4 | 3 (15) |
| | 5 | 2 (10) |
| Anyone in the household in paid employment | Yes | 17 (85) |
| | No | 3 (15) |
| Educated to degree level | Yes | 13 (65) |
| | No | 7 (35) |
| Single parent | Yes | 5 (25) |
| | No | 15 (75) |

This table was created by the coauthors for this publication.
IMD, Index of Multiple Deprivation.

## Data analysis

Data were analysed thematically[18] using NVivo (V.12) to aid data management and analysis. All transcripts were read by JC, RB and HT to gain familiarity with the data. Three transcripts were independently coded by these researchers to generate an initial list of codes. Codes were both deductive (generated from topic guide and research questions) and inductive (generated from interview data). Differences in coding were resolved through discussion to produce an agreed coding framework. Subsequent transcripts were single coded using this framework with further discussion to clarify or expand the framework as needed.

## RESULTS
### Participants

Sixteen mothers and four fathers participated in the study (table 1). Average age of parents was 34 years (range 21–45 years). Sixty percent of the sample were White British or White Other. Thirteen participants were educated to degree level. Half resided in the most deprived Index of Multiple Deprivation (IMD) quintile. Five participants were single parents. Thirteen reported at least one parent was not working during the lockdown period; three reported that no household member was currently employed. Two participants had no access to a garden. Despite the variation within our sample, we found few differences in the reported impacts of lockdown across the socioeconomic spectrum, or by other participant characteristics.

## Food and eating
### Increased snacking

Most parents reported their child's snacking had increased over lockdown, which they often linked to stay-at-home restrictions. Children were bored and consequently snacked more: '*When there's nothing to occupy them, they've got to do something*' (Mother_1). Some felt loss of routine had disrupted rules around snacking: '*if he's at pre-school he can't snack. He can only eat at the specific snack time… Here, he just thinks it's on tap*' (Mother_2). Some found dealing with demands for snacks challenging, with some parents permitting more 'unhealthy' items than normal: '*It's all we hear. She's constantly hungry… She's definitely eating more bags of crisps than she normally would*' (Mother_3).

Some parents created makeshift rules around food, balancing child demands with wanting to provide

'healthy' snacks. Mother_4's children were allowed crisps because they went '*on and on*' but then were told '*to move on to the fruit bowl*'. Another parent made healthy snacks more accessible: '*I make sure there are bits and bobs cut up in the fridge… If she's going to help herself to something, it's the stuff on the bottom she can reach*' (Mother_5). However, some parents gave additional 'treats' to their children to compensate for the pandemic's impact: '*We were feeling bad they'd missed out on so much, we were buying treats to make them feel happy*' (Mother_6).

### Food preparation and mealtimes

Some parents reported an increase in problematic meal-time behaviours, such as children not wanting to eat or taking a long time to eat. Often this was linked to increased snacking. However, several reported positive effects on mealtime routines. With parents working from home, families could eat together more often. Similarly, parents enjoyed having time to cook and involve their child. This helped occupy their child and taught them new skills: '*When we couldn't go to McDonald's we made home-made chicken nuggets… that's been posi*tive, *they've enjoyed cooking*' (Mother_6).

Lockdown had other impacts on families' food behaviours. With restaurants shut, people no longer ate out. Some participants also reported consuming less take-aways. For one, this was related to concerns about the virus: '*I didn't touch any takeaways. I didn't feel comfortable*' (Mother_4). In contrast, one parent reported eating more takeaways because '*w*e *all need something to look forward to*' (Mother_7).

### Food costs

Several participants reported spending more on food during lockdown. Some related this to the whole family being at home, as well as increased snacking, while others felt prices increased. One family no longer had access to free meals at their child's nursery. Another participant who was shielding had moved to online deliveries, a service unavailable at their normal cheaper supermarkets: '*We'd normally just go to Aldi or Lidl… now we've been trying to do the online stuff, it's just added to our bill*' (Mother_9). However, an unexpected benefit of this was healthier snacking: '*it's a money thing, we've not been able to buy all the junk food*' (Mother_9).

### Physical activity

Most parents felt their child was less active during lock-down. For some this was substantial—'*we* [were] *nowhere near as active as we used to be*' (Mother_4)—while a minority felt it had little impact. Several related this reduction to childcare closures which removed usual opportunities for activity: '*I put it down to routine and being less active… At nursery, you're on the go from the time you get there to the time you go home*' (Mother_10). Opportunities for active transport to and from childcare were also lost: '*[She's] definitely less active because we walk to* [pre-]*school and that was 10 minutes there and 10 minutes back… She's not having that exercise*'

(Mother_11). Normal activities (like swimming, soft play) were also closed and opportunities to socialise with others restricted.

Parents noted several factors that helped children stay active. Having a sibling was important, providing someone to play with. Equally, access to play equipment helped facilitate activity: '*They've got a trampoline, a bike each, a scooter each*' (Mother_4). Access to outside space was also important. Only two parents had no access to outdoor space, but both described this as challenging during lock-down. As Mother_8 explained: '*We couldn't leave and there was nothing really you could do. And living in a block of flats, you can't be too noisy because you've got neighbours everywhere.*' In contrast, Mother_1 noted how having a safe space to cycle outside her house helped keep her son active: '*My front* [garden] *is really nice for children to play and ride bikes.*'

Good weather was also seen as an important facilitator of activity: '*We had the pool out and [he] was using it quite a lot … We were pretty much burning energy off most of the time, especially with the nice weather we had*' (Mother_12). Some parents discussed how they made use of the local environment during lockdown: '*We've done a lot of exploring around the local area… we've found some places we didn't know were there*' (Mother_6).

Parents often put effort into keeping their child physically active: '*I've tried really hard to make sure he's been as active as possible*' (Mother_7). This was easier for families where one parent was not working: '*He has to have a parent with him. "Look at me! Look at my star jumps!"… That was the difficult bit, because there is only so much of that you can do, especially when you've got work*' (Father_1).

### Sedentary behaviours

Parents reported their child engaged in a range of seden-tary activities during lockdown (like reading, drawing) but screen time was mentioned most frequently, with almost all parents reporting substantial increases. For many families, screen time filled the void left by being unable to go out or socialise. Mother_13 explained they watched more television because '*we were stuck in the house… we weren't having that release of getting out*', while Mother_1 commented: '[My son] *can't go to friends, he can't meet family… so the screen had become his friend.*'

Many felt screen time had been useful during this diffi-cult period, with one parent referring to it as a '*lifesaver*' (Father_1). For those trying to work from home, it was often the only way they could manage: '*If I've got a two hour Zoom meeting and I don't want him bursting in every five seconds I'd give him his iPad and he'll watch a film*' (Mother_6). Even for those not working, managing everyday chores was a challenge: '*You've still got housework to do, washing to do, meals to prepare… You can't always be there 24/7 to keep her occupied*' (Mother_3).

Screen time was also used to provide much needed respite from the intensive parenting effort lockdown enforced. Father_2 described screen time as providing '*a break*' while Mother_11 used it to avoid constant sibling squabbles: '*They're just following you around … fighting and*

hitting each other… just sit down quietly and watch a movie.' However, parents' language suggested many felt uncomfortable about increased screen time. Mother_6 admitted: *'It's bad, I don't even like saying it out loud, but [screen time] was almost like a pacifier, really,'* while Mother_3 admitted: *'It's going to sound horrible, but it's just sometimes easier.'*

Parents often distinguished between 'good' and 'bad' screen time. Educational or interactive screen time was better than passive television watching. Mother_11 would ask herself: *'Is it educational or is it just rubbish they're watching?',* while Mother_3 explained: *'I don't want her to mindlessly sit there watching a screen. At least if she's playing [educational] games, it's making her think.'*

Concerns over screen time were rarely linked to lack of movement or activity. Rather, Father_3 referred to *'the effect of the brain becoming lazy'.* Mother_5 suggested too much screen time interfered with normal childhood and fostered an inability to occupy oneself: *'They're not being children. I suppose you think about your own childhood, don't you? All the bits and bobs I used to do as a kid. When I've said no screen time, they just wander around like aimless sheep.'*

Some parents created new rules to manage increased screen time. This included restrictions on the amount, times of day or content. Mother_14 explained, *'We have had to impose rules… we've just had to structure it a bit more, because she would gladly sit in front of the telly or iPad all day.'* Other families took a more reactive approach: *'We're not a routine sort of family anyway, so if they've had an hour or so, then I'll be like, "Right, that's it now, everybody out"'* (Mother_5).

### Sleep

Almost half of parents reported negative changes to their child's sleep. Many reported difficulties in getting their child to sleep, with some staying up very late. Parents often related this either to lack of activity—*'it's because she's not been as active and been out'* (Mother_8)—or increased screen time—*'a direct correlation of being a bit more bored, spending more time on a screen'* (Mother_3). Loss of routine was again a source of difficulty for some families. As Mother_4 commented: *'Before it was a very strict* [bedtime] *routine … now it is kind of as and when.'* A minority of parents wondered if anxiety played a part in their child's sleep difficulties: *'I'm just wondering whether* [his frequent waking] *was a part of maybe anxiety or worried about everything that's going on'* (Mother_12).

### Longer term impacts

Generally, parents thought children's increase in snacking during lockdown was *'just a temporary thing'* (Mother_2). As Mother_5 acknowledged: *'when she goes to school she's not going to have that opportunity to (a) steal food or (b) she won't be home to be eating masses during the day.'* However, one parent whose child had already returned to nursery acknowledged getting back into a routine was difficult: *'Trying to get him to eat properly once we started to go back into a routine was really hard'* (Mother_10).

Several parents wished to sustain positive changes established during lockdown, such as home-cooked meals and involving children in food preparation. However, these changes required time and would need a *'conscious effort to keep doing'* (Mother_14). Many talked positively about returning to previous activities (eg, swimming, gymnastics) though several admitted feeling *'wary'* (Father_4) about how safe these were, especially those involving close contact.

Some parents were concerned about the impact of lockdown on their child's skills and confidence. Mother_14 felt her daughter had lost her swimming skills and noted a general loss of confidence: *'She's a lot more cautious on her bike because for a while they just didn't leave the house… she's had a step backwards with that.'* Another was concerned about lost stamina and wondered if her daughter would manage the mile-long walk to school: *'I am actually a little bit worried about how she's going to fare… she's not been used to doing that walk'* (Mother_5).

Most parents were not overly concerned about increased screen time, assuming it would naturally decrease with a return to routine. However, others wondered how an increased reliance on screens during lockdown might change their child's behaviours going forward. Mother_6 described her son before lockdown as a *'pocket rocket'* who never sat still. Now, however, *'rather than going outside and running around the garden, as soon as he wakes up [he asks] "Can I have the iPad?"…He would quite happily sit all morning'.* Similarly, Mother_3 explained her daughter would now *'rather have something on the telly than go out'.* Allowing her daughter to watch television in her mother's bedroom during lockdown, Mother_3 acknowledged the difficulty in re-establishing prelockdown boundaries: *'She's asking Santa for a television for her bedroom, which I'm hoping she forgets.'*

### DISCUSSION

During the first COVID-19 lockdown parents reported increased snacking, screen time and sleep issues, and decreased physical activity in their preschool children. There were also positive family behaviour changes, such as more eating together and use of the local environment for physical activity. Some parents expressed concerns about the future impact of negative behaviour changes, but most believed these were a temporary consequence of a lack of routine.

This study provides in-depth reports of family life with a preschool child during the pandemic, and complements findings of parents of 2–4 year-olds from the Co-SPYCE survey[15] which revealed children's screen time as a key parental stressor during lockdown. Parental use of children's screen time to 'get things done' was a key finding of a 2017 qualitative systematic review.[19] In our study, with the absence of childcare, and corresponding increase in parental juggling of work and household chores, it is clear how this 'need' for downtime increased for many.

Most parents felt their child was less active over lockdown echoing findings from other studies. Only 22% of 2–4 year-olds in the Co-SPYCE study were reported

as getting 180 min of activity per day[15] and almost half (47%) of Scottish parents reported their children (2–7 years) did less physical activity during lockdown.[16] A Canadian survey of parents of young children (18 months to 5 years) reported half of children were less physically active and 87% were engaged in more screen time during the initial COVID-19 restrictions compared with prepandemic.[20] Surveys of older children in the USA and Canada found children were less active and engaged in more screen time during the initial COVID-19 outbreak compared with before restrictions.[21 22] Parent encouragement, participation and support of physical activity were found to be important correlates of children's physical activity.[22] Zecevic et al[23] showed that, in circumstances not associated with lockdown, young children who receive parental support for physical activity were more likely to be highly active than inactive. In our study, some parents made considerable efforts to keep their child active using the opportunity to leave the house once per day for physical activity to explore the local environment (in contrast to restrictions in other countries such as Spain where the population was not permitted to leave home for exercise[24]). Yet despite these parental efforts, barriers specific to lockdown were reported which reduced activity, mainly relating to the closure of childcare and playgrounds, and reduced contact with friends and family.

Sleep disruptions reported by parents in our study echo findings of a questionnaire study with mothers of Italian preschoolers where more challenging bedtime routines and decreased sleep quality were reported as an early impact of lockdown.[25]

Our study identified disruptions to normal eating habits and routines including changes in snacking behaviour, frequency and volume of eating and who children were eating with. While some families speculated these changes would soon return to normal, it may be difficult to re-establish boundaries and break new habits.[26 27] One parent discussed how free meals for her child at nursery had ceased. While our study did not identify acute food poverty, it is likely that preschool children will have experienced food poverty where free school meals were no longer available and where the pandemic impacted family finances.[28]

Rundle et al[29] suggested the closure of schools during this period may exacerbate childhood obesity levels. While few parents mentioned weight, our findings suggest there is potential for obesity prevalence in preschool children to have increased due to increased snacking and screen time, and decreased physical activity, particularly if changes during lockdown have persisted. The annual National Child Measurement Programme was stopped in March 2020 but restarted in January 2021. This will enable assessment of the impact on children who started school in September 2020.[30]

The negative impacts of COVID-19 are expected to be greatest for children living in poverty.[31 32] We sought to explore differences in the impact of lockdown across the socioeconomic spectrum. Half of our participants lived in the most deprived IMD quintile, giving insight into those living in the poorest areas. We noted some differences within our sample. Two families without access to gardens reflected on the difficulty in keeping children active. Families where parents were not working found it easier to engage children in physical activity. Beyond this, however, we did not note any obvious differences between those living in more or less deprived neighbourhoods. The impact of lockdown on children's behaviours was felt similarly across the socioeconomic spectrum. The lack of difference could be due to sample characteristics; most families still had at least one parent working and access to a garden. Interviews took place early in the COVID-19 crisis, so we may not yet be seeing the full effect, for example, with the protective effect of the furlough scheme still in place.

This study raises questions about how parents can be supported during future lockdowns or local restrictions. Subsequent UK lockdowns in late 2020 and early 2021 presented new and additional challenges: shorter, colder days; weariness of the restrictions; and increased financial pressures with high rates of unemployment and redundancy.[33] In addition, positive cases in childcare or school settings at this time could result in children being required to isolate for 10 days without being allowed out. The importance of supporting families to maintain or increase activity for young children within the home and in accessible outdoor environments becomes more prominent. Families need support to establish healthy revised routines and manageable healthy rules for snacks and screen time during periods of restriction. This must be presented in a context of supporting parents and not adding guilt or burden to parents during a period of stress. A recent Ipsos MORI report, which included an online survey of 1000 parents of 0–5 year-olds, found 70% of parents felt judged by others and almost half felt this negatively impacted on their mental health.[34] There is a role here for all sectors of health, social care and education including health visitors, children's centres, early years providers, general practitioners, social services, local authority public health teams and the new Office for Health Improvement & Disparities. Further, there is a role for research with preschoolers and 2020 school starters to be expanded[35] to understand the impact on their physical and social development, and particularly their risk of obesity, as they start school.

### Strengths and limitations

In-depth interviews enabled us to gain novel insights into the experiences of parents and preschool children during lockdown and restrictions, though we acknowledge the limitation of investigating preschool children's experiences by asking their parents, rather than involving preschool children directly. A strength of the study was the ethnic and socioeconomic diversity of the sample from two regions of England, although this was a relatively small (n=20) and educated sample. Parents volunteering for interview may have had more interest in the topic or

had different experiences than those who did not volunteer. We acknowledge that the use of remote interviews could have compromised participants' ability to express themselves and non-verbal clues may have been missed.[36] Other primary caregivers were eligible for the study, but none were recruited. Rigour was achieved by detailed data analysis and analytical decisions being shared with all team members to achieve credibility.

## CONCLUSIONS

While some positive changes were reported, there were widespread examples of lack of routines, habits and boundaries which, at least in the short term, were likely to have been detrimental for child health and development. Guidance and support for parents and families on how to maintain healthy routines and compensate for future COVID-19 restrictions could be valuable to protect child health and ensure that unhealthy habits are not adopted.

**Correction notice** This article has been corrected since it was published. The term 'White' has been changed to 'White British or White Other'. The funding statement has also been updated.

**Acknowledgements** We are very grateful to all the parents who took part in the research interviews.

**Contributors** The study was conceived by JC, RK, SC, KW, HT, RB, KH, SAS and RL. JC led the study with oversight from RL and RK. Interviews were conducted by JC, SC and KW. Coding of the data was performed by JC, HT and RB. JC, RK and RL produced the first draft of the manuscript, with all other authors providing critical review and intellectual content. All authors read and approved the final manuscript.

**Funding** This work was funded by the NIHR School for Public Health Research (SPHR-PROG-CYP-WP3) and NIHR funding for the NAP SACC UK trial (2019-3426). SAS and SC were supported by the Medical Research Council and the Chief Scientist Office of the Scottish Government Health and Social Care Directorates (MC_UU_00022/1 and SPHSU16).

**Disclaimer** The views expressed in this publication are those of the authors and not necessarily any of the funding bodies listed.

**Competing interests** None declared.

**Patient consent for publication** Not required.

**Ethics approval** Ethical approval for the study was granted by the University of Bristol Faculty of Health Sciences Research Ethics Committee (ref: 106002). Participant consent was recorded on a separate audio file before the interview commenced.

**Provenance and peer review** Not commissioned; externally peer reviewed.

**Data availability statement** Data are available upon reasonable request. Anonymised study data will be made available via a University of Bristol repository.

**ORCID iDs**
Joanne Clarke http://orcid.org/0000-0003-2563-5451
Rebecca Langford http://orcid.org/0000-0002-7722-0808

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
