## [Reviewer comments · BMJ Open]

ARTICLE DETAILS

TITLE (PROVISIONAL)	Impact of COVID-19 restrictions on pre-school children's eating, activity and sleep behaviours: a qualitative study
AUTHORS	Clarke, Joanne; Kipping, Ruth; Chambers, Stephanie; Willis, Kate; Taylor, Hilary; Brophy, Rachel; Hannam, Kimberly; Simpson, Sharon; Langford, Rebecca

VERSION 1 – REVIEW

REVIEWER	Visram, Shelina Newcastle University
REVIEW RETURNED	16-Apr-2021

GENERAL COMMENTS	BMJ Open review (April 2021) Impact of COVID-19 restrictions on pre-school children's eating, activity and sleep behaviours: a qualitative study Thank you for the opportunity to review this manuscript, which describes a qualitative study to explore parents' views on the impact of the first UK lockdown on their pre-school children. I enjoyed reading the manuscript, as both a public health researcher and parent of a pre-school child myself. The findings are important in terms of describing short-term changes in children's health-related behaviours that could have longer-term effects on health and wellbeing (although the emphasis is on risk of developing obesity). It would have been interesting to repeat the study during or after the second and third national lockdowns, to see whether and how families had adapted to subsequent COVID-19 restrictions. The study focuses exclusively on parents' views as opposed to other primary caregivers; this could have been acknowledged as a further limitation of the study. My other suggested revisions are relatively minor: 1. Abstract:o Page 2, line 41: 'the COVID-19 lockdown' implies a singular lockdown; maybe re-word to 'the first COVID-19 national lockdown'?
--

	 ○ Page 3, line 6: will all readers know what is meant by ‘the most deprived quintile’? This could be changed to ‘those living in the 20% most deprived areas’ or ‘the most deprived areas’. ○ Line 43 and 48: I would prefer ‘reported impact’ or ‘perceived impact’ as changes in behaviour and diet were not measured in any way. ○ Page 4, line 1: Missing word; ‘conducted with parents’. 2. Aim: The research aim/question/objective is not clearly stated. The final paragraph of the background section could be re-worded to make clear the specific gap in existing knowledge this study was designed to fill. The statement beginning with ‘We explored...’ would be better worded as ‘The aim of this study was to explore...’. 3. Methods: I did not think it was appropriate to describe consultation with nursery staff as PPI. Instead, I suggest removing this sub-heading and mentioning that nursery staff provided advice on recruitment under the recruitment sub-heading. In the data analysis sub-section, what is meant by ‘Transcripts were read by JC/RB/HT to gain familiarity with the data’? I was not clear on whether this meant that all three researchers read all of the transcripts or that each transcript was read by one of three researchers. 4. Results: In the discussion you mention that there were few differences in the impact of lockdown across the socioeconomic spectrum. This would be better stated at the start of the results section so that the reader understands why findings/quotes are not attributed to participants from particular socioeconomic backgrounds (or genders, child ages, etc – presumably there were no differences by these characteristics either). On a final note, I was surprised to see that the interview topic guide (table 1) was comprised largely of yes/no answer questions. There were also no questions on differences in experiences or behaviours during and after lockdown. I appreciate that the interviews were conducted after the first lockdown had ended but the stated ‘aim’ included exploring the impact of lockdown and its easing. There is no specific revision being requested here; I am just unsure how this aim was fulfilled using the closed questions shown in table 1.
--	---

REVIEWER	Proulx, Kerrie Lunenfeld-Tanenbaum Research Institute
REVIEW RETURNED	13-May-2021

GENERAL COMMENTS	This paper discusses the effects of a COVID-19 lockdown/movement restrictions on young children's eating and sleeping habits and sedentary behaviour. The main contribution of the paper is that it provides a qualitative lens to issues that have been investigated primarily via caregiver surveys. As such, these
---

qualitative results are a welcomed remedy to the present knowledge gap.

I have some remarks that the authors may wish to address to further improve the manuscript and to acknowledge some limitations:

1. Interview data were collected over the phone/remotely at one point in time with a relatively small sample of parents, limiting in-depth analysis and firm conclusions based on the study design. While this cannot be rectified at this time, the authors could take a more modest view on the study design (i.e., rapid vs. in-depth qualitative methods) and include methodological features (i.e. sample size, data collection methods) as potential study limitations.

2. In addition to Co-SPYCE, there are numerous studies that have found increased screen time, sleep disturbance and more sedentary behaviour among children during the COVID-19 pandemic. Some studies focus specifically on young children while others focus on childhood more broadly. I would encourage the authors to include a more complete review of the existing literature on this topic and discuss how their findings compare/contrast and add value to what has already been published.

References that may be useful for consideration:

Carroll N, Sadowski A, Laila A, et al. The Impact of COVID-19 on Health Behavior, Stress, Financial and Food Security among Middle to High Income Canadian Families with Young Children. *Nutrients* 2020;12. doi:10.3390/nu12082352

Dunton GF, Do B, Wang SD. Early effects of the COVID-19 pandemic on physical activity and sedentary behavior in children living in the U.S. *BMC Public Health* 2020;20:1351. doi:10.1186/s12889-020-09429-3

Moore SA, Faulkner G, Rhodes RE, et al. Impact of the COVID-19 virus outbreak on movement and play behaviours of Canadian children and youth: a national survey. *Int J Behav Nutr Phys Act* 2020;17:85. doi:10.1186/s12966-020-00987-8

Idoiaga Mondragon N, Berasategi Sancho N, Dosil Santamaria M, et al. Struggling to breathe: a qualitative study of children's wellbeing during lockdown in Spain. *Psychol Health* 2020;:1–16. doi:10.1080/08870446.2020.1804570

Dellagiulia A, Lionetti F, Fasolo M, Verderame C, Sperati A, Alessandri G. Early impact of COVID-19 lockdown on children's sleep: a 4-week longitudinal study. *J Clin Sleep Med*. 2020;16(9):1639–1640

3. The text on page 5 refers to lockdown impacts on sedentary behaviour and sleep patterns but the references (7-8) are from earlier periods. Consider checking these references for accuracy.

	4. The discussion points towards differences in experiences based on socio-economic circumstances and access to outdoor space. Since both single and coupled parents are included in this study. Given limited evidence on the topic, I would urge the authors to consider including analysis and discussion of similarities or differences in caregiving perspectives/experiences during lockdown among single vs. coupled parents, if such data are available.
--	---

VERSION 1 – AUTHOR RESPONSE

Reviewer 1 comment	Authors' response
Thank you for the opportunity to review this manuscript, which describes a qualitative study to explore parents' views on the impact of the first UK lockdown on their pre-school children. I enjoyed reading the manuscript, as both a public health researcher and parent of a pre-school child myself. The findings are important in terms of describing short-term changes in children's health-related behaviours that could have longer-term effects on health and wellbeing (although the emphasis is on risk of developing obesity). It would have been interesting to repeat the study during or after the second and third national lockdowns, to see whether and how families had adapted to subsequent COVID-19 restrictions. The study focuses exclusively on parents' views as opposed to other primary caregivers; this could have been acknowledged as a further limitation of the study.	Thank you for your comments. We are pleased that you enjoyed reading the manuscript. We agree that it would have been interesting to repeat the study during or after the second and third national lockdowns. Unfortunately, this was not possible due to time and budgetary constraints. Other primary caregivers were eligible to participate in the study (as outlined in the criteria for inclusion (page 6, line 50 – 'parent/carer of a child aged 3-5 years'). However, we did not have any other primary caregivers come forward to participate in the study. We have now acknowledged this in the limitations: "Other primary caregivers were eligible for the study, but none were recruited." (Page 16, lines 298-9)
Abstract: - Page 2, line 41: 'the COVID-19 lockdown' implies a singular lockdown; maybe reword to 'the first COVID-19 national lockdown'?	Changed as suggested.
Abstract: Page 3, line 6: will all readers know what is meant by 'the most deprived quintile'? This could be changed to 'those living in the 20% most deprived areas' or 'the most deprived areas'.	Changed to 'the most deprived areas'.
Abstract: Line 43 and 48: I would prefer 'reported impact' or 'perceived impact' as changes in behaviour and diet were not measured in any way.	We have now made the following changes, as suggested: Abstract: "Parents reported that lockdown negatively impacted on pre-school children's eating, activity and sleep routines." Strengths and Limitations: "Few studies have explored the perceived impact of COVID-19 restrictions on young children or health behaviours like physical activity or diet."

	“Our in-depth interviews provide novel insights into the perceived impact of lockdown restrictions on pre-school children’s physical activity, sedentary behaviour, food intake and sleep.”
Page 4, line 1: Missing word; ‘conducted with parents’.	Changed as suggested.
Aim: The research aim/question/objective is not clearly stated. The final paragraph of the background section could be re-worded to make clear the specific gap in existing knowledge this study was designed to fill.	Thank you. The Background section has been re-ordered and the specific gap in existing knowledge has been added explicitly: ‘While these quantitative surveys provide a useful overview, few studies have used qualitative methods to understand in greater depth how pre-school children have been affected by these restrictions. To address this gap, we used in-depth interviews with parents of 3-5 year-olds to explore the impact of the first UK COVID-19 lockdown on pre-school children’s eating, activity and sleep behaviours.’ (Page 5, lines 37-40)
The statement beginning with ‘We explored...’ would be better worded as ‘The aim of this study was to explore...’.	As noted above, we have rephrased sentence to identify the specific research gap we were addressing.
Methods: I did not think it was appropriate to describe consultation with nursery staff as PPI. Instead, I suggest removing this sub-heading and mentioning that nursery staff provided advice on recruitment under the recruitment sub-heading.	This is a required sub-heading for the journal so we have not removed it but have provided the following clarification: “Due to time constraints, we did not directly involve parents or children in the planning of this study. However, nursery staff provided advice on the best ways to recruit parents for this study.” (Page 6, lines 59-60)
Methods: In the data analysis sub-section, what is meant by ‘Transcripts were read by JC/RB/HT to gain familiarity with the data’? I was not clear on whether this meant that all three researchers read all of the transcripts or that each transcript was read by one of three researchers.	This has been clarified and now reads: ‘All transcripts were read by JC, RB and HT to gain familiarity with the data’. (Page 6, line 70)
Results: In the discussion you mention that there were few differences in the impact of lockdown across the socioeconomic spectrum. This would be better stated at the start of the results section so that the reader understands why findings/quotes are not attributed to participants from particular socioeconomic backgrounds (or genders, child ages, etc – presumably there were no differences by these characteristics either).	Sentence added to the start of the results section: ‘Despite the variation within our sample, we found few differences in the reported impact of lockdown across the socioeconomic spectrum, or by other participant characteristics.’ (Page 7, lines 86-7)

I was surprised to see that the interview topic guide (table 1) was comprised largely of yes/no answer questions. There were also no questions on differences in experiences or behaviours during and after lockdown. I appreciate that the interviews were conducted after the first lockdown had ended but the stated 'aim' included exploring the impact of lockdown and its easing. There is no specific revision being requested here; I am just unsure how this aim was fulfilled using the closed questions shown in table 1.	Thank you for picking this up. This is our mistake as we failed to indicate that this was a summary. For clarity, we have now revised the table to highlight the topic areas covered and provide the full topic guide as supplementary material. This is indicated in the table legend as well as in the methods section (page 6, line 70).
---	---

Reviewer 2 comment	Authors' response
This paper discusses the effects of a COVID-19 lockdown/movement restrictions on young children's eating and sleeping habits and sedentary behaviour. The main contribution of the paper is that it provides a qualitative lens to issues that have been investigated primarily via caregiver surveys. As such, these qualitative results are a welcomed remedy to the present knowledge gap.	Thank you for your comments.
Interview data were collected over the phone/remotely at one point in time with a relatively small sample of parents, limiting in-depth analysis and firm conclusions based on the study design. While this cannot be rectified at this time, the authors could take a more modest view on the study design (i.e., rapid vs. in-depth qualitative methods) and include methodological features (i.e. sample size, data collection methods) as potential study limitations.	We agree these limitations need to be acknowledged and have added the sample size and remote data collection methods as limitations of the study. The conduct and analysis of these interviews was rigorous and systematic. Having added in these limitations, we feel we have been suitably circumspect in our conclusions and have acknowledge the need for further work in this area (page 16, lines 292-4). 'A strength of the study was the ethnic and socioeconomic diversity of the sample from two regions of England, although this was a relatively small (n=20) and educated sample.' (Page 16, line 295) 'We acknowledge that the use of remote interviews could have compromised participants' ability to express themselves and non-verbal clues may have been missed.' (Page 16, lines 297-8)
In addition to Co-SPYCE, there are numerous studies that have found increased screen time, sleep disturbance and more sedentary behaviour among children during the COVID-19 pandemic. Some studies focus specifically on young children while others focus on childhood more broadly. I would encourage the authors to include a more complete review of the existing literature on this topic and discuss how their	Thank you for these useful suggestions. The paper by Moore et al. was already included within our discussion section. The other suggested references have been included to improve our discussion (Pages 13-14).

findings compare/contrast and add value to what has already been published. References that may be useful for consideration: Carroll N, Sadowski A, Laila A, et al. The Impact of COVID-19 on Health Behavior, Stress, Financial and Food Security among Middle to High Income Canadian Families with Young Children. Nutrients 2020;12. doi:10.3390/nu12082352 Dunton GF, Do B, Wang SD. Early effects of the COVID-19 pandemic on physical activity and sedentary behavior in children living in the U.S. BMC Public Health 2020;20:1351. doi:10.1186/s12889-020-09429-3 Moore SA, Faulkner G, Rhodes RE, et al. Impact of the COVID-19 virus outbreak on movement and play behaviours of Canadian children and youth: a national survey. Int J Behav Nutr Phys Act 2020;17:85. doi:10.1186/s12966-020-00987-8 Idoiaga Mondragon N, Berasategi Sancho N, Dosil Santamaria M, et al. Struggling to breathe: a qualitative study of children's wellbeing during lockdown in Spain. Psychol Health 2020;:1–16. doi:10.1080/08870446.2020.1804570 Dellagiulia A, Lionetti F, Fasolo M, Verderame C, Sperati A, Alessandri G. Early impact of COVID-19 lockdown on children's sleep: a 4-week longitudinal study. J Clin Sleep Med. 2020;16(9):1639–1640	
The text on page 5 refers to lockdown impacts on sedentary behaviour and sleep patterns but the references (7-8) are from earlier periods. Consider checking these references for accuracy.	We have now clarified this text so it is clear that we are relating the impact on physical activity, sedentary behaviours and sleep patterns with the closure of pre-schools. 'With the closure of pre-schools, lockdown reduced children's opportunities for physical activity, while increasing the likelihood of sedentary behaviours [7] and had the potential to disrupt sleep patterns [8].' (Page 4, lines 16-7)
The discussion points towards differences in experiences based on socio-economic circumstances and access to outdoor space. Since both single and coupled parents are included in this study. Given limited evidence on the topic, I would urge the authors to consider including analysis and discussion of similarities or differences in caregiving	Thank you for this suggestion. However, having re-visited the data, we were unable to establish any patterns when comparing single vs. coupled parents with respect to eating and activity behaviours.

perspectives/experiences during lockdown among single vs. coupled parents, if such data are available.	
--	--

VERSION 2 – REVIEW

REVIEWER	Visram, Shelina Newcastle University
REVIEW RETURNED	30-Jul-2021

GENERAL COMMENTS	All necessary revisions have been made and so I am happy to recommend that this manuscript be accepted for publication. However, if there is the opportunity to make further minor revisions, I would suggest the following: Abstract: 'Reported' is used three times in the first two sentences of the conclusion and so you may want to re-word. Line 17: 'With the closure of pre-schools...' could also mention the closure of playgrounds and other spaces for young children to play and be active. Line 234 - 'Surveys' (plural).
--

REVIEWER	Proulx, Kerrie Lunenfeld-Tanenbaum Research Institute
REVIEW RETURNED	25-Aug-2021

GENERAL COMMENTS	Thank you for addressing all the comments. The revisions have contributed to an improved version of the manuscript, which I feel is ready for publication and will provide valuable information for readers.
--

VERSION 2 – AUTHOR RESPONSE

Reviewer: 1

Dr. Shelina Visram, Newcastle University

Comments to the Author:

All necessary revisions have been made and so I am happy to recommend that this manuscript be accepted for publication. However, if there is the opportunity to make further minor revisions, I would suggest the following:

Abstract: 'Reported' is used three times in the first two sentences of the conclusion and so you may want to re-word.

Thank you for identifying this. We have now revised the text as follows:

“Parents reported that lockdown negatively impacted on pre-school children's eating, activity and sleep routines. While some positive changes were identified, many participants described lack of routines, habits and boundaries which may have been detrimental for child health and development.”

Line 17: 'With the closure of pre-schools...' could also mention the closure of playgrounds and other spaces for young children to play and be active.

We agree this needed to be expanded and have made the following revision:

“The closure of pre-schools, playgrounds and the ‘stay at home’ order during lockdown reduced children’s opportunities for physical activity, increased the likelihood of sedentary behaviours [7] and had the potential to disrupt sleep patterns [8].”

Line 234 - 'Surveys' (plural).

This typo has been corrected.

Reviewer: 2

Dr. Kerrie Proulx, Lunenfeld-Tanenbaum Research Institute

Comments to the Author:

Thank you for addressing all the comments. The revisions have contributed to an improved version of the manuscript, which I feel is ready for publication and will provide valuable information for readers.

Thank you.